# Trends of Medical Service Utilization for Tinnitus: Analysis Using 2010–2018 Health Insurance Review and Assessment Service National Patient Sample Data

**DOI:** 10.3390/healthcare10081547

**Published:** 2022-08-16

**Authors:** Taewoon Min, Jiyoon Yeo, Ye-Seul Lee, Song-Yi Kim, Donghyo Lee, In-Hyuk Ha

**Affiliations:** 1Jaseng Hospital of Korean Medicine, 536 Gangnam-daero, Gangnam-gu, Seoul 06110, Korea; 2Department of Economics, Korea University, 145 Anam-ro, Seongbuk-gu, Seoul 02841, Korea; 3Jaseng Spine and Joint Research Institute, Jaseng Medical Foundation, 3F, 538 Gangnam-daero, Gangnam-gu, Seoul 06110, Korea; 4Department of Acupoint and Anatomy, College of Korean Medicine, Gachon University, Seongnam 13120, Korea; 5Department of Ophthalmology, Otolaryngology, and Dermatology, College of Korean Medicine, Woo-Suk University, Jeonju 55338, Korea

**Keywords:** tinnitus, Korea, medical expenditure

## Abstract

Given the increasing prevalence of tinnitus and expenditure related to its treatment, it is important to identify the efficacy of different treatment methods used for its diagnosis and treatment. To this end, this study analyzed the trends of medical service utilization for tinnitus in adult patients from 2010 to 2018 based on a national sample of medical claims data from the Health Insurance Review and Assessment Service National Patient Samples database. A total of 94,323 patients with tinnitus were identified in Korea between 2010 and 2018. The results confirmed that the number of patients, claim numbers, and expenditures steadily increased during the nine-year period. Blood circulation agents were the most commonly used drug therapy; however, the frequency of their use gradually decreased, whereas that of tinnitus and vertigo medicines gradually increased. Total and average expenditure per patient nearly doubled in this period. The study showed that medication trends are changing from blood circulation agents to tinnitus or vertigo medicines. The findings of this study may be helpful for clinicians and researchers in the study, treatment, and management of tinnitus.

## 1. Introduction

Tinnitus is defined as a phantom perception of sound without any external stimulation. The symptoms of tinnitus can generally be described as a ringing in the ear; its location may be perceived as either inside or outside of the head and unilateral in most cases, although it can also be perceived as bilateral in some cases. The sound perceptions include humming, ringing, hissing, and cicada sounds [1]. The experience of tinnitus that people commonly refer to concerns subjective tinnitus, which only the patient can hear. Although the exact cause of tinnitus is unknown, it may be due to a malfunction of the auditory end organs, commonly caused by aging-related degeneration. Other causes include Meniere’s disease, trauma, and cardiovascular disease (CVD) [2]. Tinnitus is reported to be linked with hearing loss, although the mechanisms underlying its association are unknown [3]. Due to its unknown etiology, a wide range of concomitant diseases, and its influence on the quality of life, some studies defined tinnitus as an independent disorder that requires a comprehensive diagnosis through etiology and patients’ comorbidities [4,5].

The prevalence of tinnitus in adult populations of various nationalities has been reported to range from 10–25% of the total population, and is gradually increasing. In Korea, the prevalence of tinnitus is reported to be 20.7%, with a higher prevalence and severity with increasing age [6,7]. However, the incidence of tinnitus has increased among younger age groups in the past decade, which may be attributed to a greater frequency of exposure to recreational noise [7,8]. Factors closely associated with tinnitus include exposure to noise, stress, and depression [7,9]; however, the association between tinnitus and variables such as sex, drinking status, education level, or income level remains unclear [9]. A high body mass index (BMI) or CVD, such as hypertension, diabetes, stroke, and angina, have been established as potential risk factors for tinnitus [7,9,10].

The management and treatment of tinnitus require significant healthcare expenditures. Moreover, according to recent studies, the average annual healthcare expenditure per patient was estimated to be approximately $2110 in the U.S. in 2014 [11], €1544 in the Netherlands in 2013 [12], and £717 in the U.K. in 2017 [13]. In the 2010 National Health Insurance Statistical Yearbook of Korea, 280,389 patients were treated for tinnitus (ICD code: H93.1), and the total estimated expenditure was 14.3 billion KRW (12.5 million USD), which then increased to 325,466 patients and 28.7 billion KRW (25.7 million USD) in 2018 [14].

Drug therapy is commonly used to treat tinnitus. It includes using blood circulation agents, antidepressants, anxiolytics, antiepileptics, and supplements such as herbal or Ginkgo biloba extracts and vitamins [1]. Though multiple studies have examined Ginkgo biloba extract’s effectiveness in treating tinnitus [15,16], the evidence of medicine used in its treatment is low [17,18,19]. For example, a Cochrane study reported a lack of evidence regarding the efficacy of antidepressants in treating tinnitus [20]. A common non-drug therapy for tinnitus is acupuncture therapy, a form of complementary and alternative medicine (CAM) therapy [21]. Other non-drug treatments include acoustic stimulation, cognitive behavioral therapy, and repetitive transcranial magnetic stimulation (rTMS) [11,17,22,23,24].

In Korea, acupuncture and herbal medications are practiced and prescribed by Doctors of Korean Medicine as the dual healthcare system divides the medical practice sectors into Western Medicine (WM) and Korean Medicine (KM). For musculoskeletal disorders and diseases with unspecified causes, KM treatments can be an option [25]. However, the exact utilization of both WM and KM has not yet been identified for tinnitus.

Owing to the increasing prevalence of tinnitus and the associated medical expenditures, the identification of robust improvements in the diagnosis and treatment of tinnitus is likely to be very important. However, few studies have explored the effectiveness of the treatment provided for tinnitus and the associated costs. Accordingly, this study aimed to analyze Health Insurance Review and Assessment Service (HIRA) claims for a nine-year period from 2010–2018 to determine the prevalence of tinnitus, related treatment details, and the usage of WM and KM in the treatment of tinnitus.

## 2. Materials and Methods

### 2.1. Data Source

This study used data from the 2010–2018 HIRA-National Patient Samples (HIRA-NPS) database. Health insurance claims data are generated when a medical institution submits a claim to HIRA after providing medical services to a patient. To increase data accessibility and convenience for researchers, HIRA conducts stratified systematic sampling of the patients in its claims database by sex and age group (in 10-year increments) and grants permission to use such data solely for research purposes. The samples provided by HIRA consist of secondary data from which personal and corporate identifying information has been removed and include treatment and prescription details for medical services claimed in the applicable year beginning with the commencement of medical services.

### 2.2. Study Design and Population

The sole inclusion criterion for this study was that a patient had received at least one type of treatment for a primary diagnosis of tinnitus (ICD-10 code H93.1) at a WM or KM institution during the applicable period. The exclusion criteria were as follows: form code indicated dentistry, health center, or psychiatry service; type of institution was a nursing hospital, psychiatric hospital, dental hospital, postpartum care facility, or health center; and total expenditure or number of days in medical care was 0 or blank (Figure 1).

### 2.3. Statistical Analysis

Patients with tinnitus were then classified by age (eight groups: 10-year increments from <15 to ≥75 years), sex, treatment payment type (national health insurance, Medicaid, and others), type of visit (inpatient and outpatient), and type of medical institution (tertiary general hospital/general hospital/hospital, clinic, KM hospital, and KM clinic), and the frequency distribution of each classification was analyzed.

The total expenditure, average expenditure, and the average number of cases were calculated to investigate changes in tinnitus treatments over the nine-year period (2010–2018). The average annual log changes for all claims by claim type and the difference between WM and KM were also analyzed.

As per the classification procedure used for HIRA medical benefits, total expenditure was divided into seven expenditure types for WM: auditory function test, other tests, injections, treatment/procedure, basic blood test/urinalysis, management/supervision, and other services, and 11 expenditure types for KM: acupuncture, electroacupuncture stimulation, dry/bloodletting cupping, direct/indirect moxibustion, hot/cold meridian therapy, infrared therapy/hot pack therapy, herbal formulation, KM tests, and other treatments. Claim frequency and expenditures were analyzed by expenditure type. Surgery, physical therapy, and diagnostic radiology were excluded as they are not covered by insurance as KM benefits.

Drugs prescribed to WM inpatients and outpatients were categorized according to the frequency of use for tinnitus based on the Anatomical Therapeutic Chemical Classification (ATC) code. Items with <0.1% in all nine years were excluded from the analysis.

All expenditures were converted using the currency exchange rate (KRW to USD) for the applicable year and adjusted based on the health sector consumer price index for 2018 (Appendix A). All statistical analyses were performed using SAS 9.4 (2002–2012 by SAS Institute Inc., Cary, NC, USA).

### 2.4. Ethical Considerations

This study protocol was approved by the public data provision deliberation committee in the HIRA and was conducted according to the relevant guidelines and regulations. The study was reviewed and qualified as an exemption by the Institutional Review Board of Jaseng Hospital of Korean Medicine, Seoul, Korea (2022-01-002). The principles expressed in the Declaration of Helsinki have been adhered to in the analysis. As the study analyzed publicly available data, no consent was obtained from the participants; moreover, the National Health Insurance Service de-identified all personal information before public release.

## 3. Results

### 3.1. General Characteristics of Medical Service Use by Patients with Tinnitus in Korea

A total of 310,098 claim records from 94,323 patients were included in the analysis. Table 1 shows that the number of patients visiting a medical institution for tinnitus was 10,460 (WM = 8908, 85%; KM = 1552, 15%) in 2010, which steadily increased each year to 11,822 (WM = 10,098, 85%; KM = 1724, 15%) in 2018. The number of patients visiting a WM institution increased slightly more than that of a KM institution. The total number of cases steadily increased from 32,791 (WM = 21,009, 64%; KM = 11,782, 36%) in 2010 to 37,744 (WM = 23,815, 63%; KM = 13,929, 37%) in 2018. The total expenditure for treatment significantly increased from $552,801.32 (WM = $390,276.5, 71%; KM = $162,524.78, 29%) in 2010 to $1,110,783.95 (WM = $819,945.73, 74%; KM = $290,838.22, 26%) in 2018. The total expenditure increased consistently during the nine-year period, except in 2013. Similarly, the average expenditure per patient steadily increased from $52.85 in 2010 to $93.96 in 2018 for all types of medical institutions (Figure 2).

### 3.2. Basic Characteristics of Patients with Tinnitus

As shown in Table 2, the age distribution of patients with Tinnitus was in the order of 55–64 years (22.68%), 65–74 years (20.6%), and 45–54 years (18.95%), indicating a high percentage of middle-aged patients. The results were similar by service type; the most common age group was 55–64 years for patients visiting WM and KM institutions, for which the percentage of patients was 22.1% and 25.67%, respectively. The overall male-to-female ratio of patients with Tinnitus was 41.87% and 58.13%, respectively, indicating a higher percentage of women. Among patients visiting WM institutions, 41.97% were men, and 58.09% were women. Among patients visiting KM institutions, 41% were men, and 59% were women. Male-to-female ratios were similar between those visiting WM and KM institutions. The results of payment types were as follows: 95.05% for National Health Insurance Service (NHIS), 4.59% for Medicaid, and 0.36% for other types.

Overall, 99.92% of medical services were outpatient treatments, and 0.08% were inpatient treatments. WM treatments took place mostly in primary care centers, with treatments at clinics accounting for 74.17% of WM services. KM treatments were more likely to occur in primary care centers, with treatments at clinics accounting for 96.12% of KM services (Appendix A).

### 3.3. Tinnitus Health Expenditure

Table 3 shows the average and total expenditures for tinnitus treatment, the total number of cases, and the rate of change over the nine-year period. For both WM and KM, the highest expenditure type was consultation fees, which had a nine-year average total expenditure of $328,970.91 with an average annual increase of 6.06%. A yearly average of 43,618.56 claims comprised consultation fees.

For WM services, testing fees were the most common claim, with a total expenditure of $249,171.13 for a total of 23,462.67 cases. This was followed by consultation fees ($246,793.19) and medication/prescription fees ($5822.39). Over the nine-year period, the highest rate of change in total expenditure was in testing fees (12.68%), and the number of cases was in hospitalization fees (9.76%). Among WM services, auditory function tests were performed most often (n = 141,201), of which tests using a pure tone audiometer (n = 66,175) were the most frequently used. Auditory brainstem response threshold tests were used in 1895 cases over a nine-year period. The average expenditure per case was $88.64, which was the highest among WM treatments; the average annual expenditure per patient was $91.09. The next highest expenditure was for blood test/urinalysis at $35.53 per patient. Considering that the average expenditure per case was $2.50, approximately 14 blood tests and/or urinalyses were performed per year (Appendix A).

Injection fees were the most common claim for KM services, with a total expenditure of $120,084.25 for a total of 36,791.67 cases. This was followed by consultation fees ($82,177.72) and medication/prescription fees ($3016.35). Over the nine-year period, the highest rates of change in both the total expenditure and the number of cases were in medication/prescription fees, with rates of change of 21.25% and 24.81%, respectively, which were substantially higher than those of the other items. Among KM treatments, acupuncture therapy was used in 203,723 cases, with an average expenditure of $3.72 per case. The average annual expenditure per patient for acupuncture therapy was $55.15, which was followed by direct moxibustion ($33.21), bloodletting cupping ($30.99), and electroacupuncture stimulation ($23.47). Among KM services, acupuncture therapy was performed an average of 15 times per patient per year, and all other treatments were performed an average of six times per patient per year (Appendix A).

### 3.4. Medication Prescribed for Patients with Tinnitus in Korea

Table 4 shows the frequency of drugs used for tinnitus. Among the therapeutics used for the treatment of tinnitus, blood circulation agents were used in the highest number of cases (n = 127,430), followed by psychiatric agents (n = 131,725) and gastrointestinal agents (n = 88,470).

Figure 3 shows the nine-year trend in the total number of drug prescriptions for tinnitus. The use of blood circulation agents gradually decreased, whereas that of tinnitus medicine and vertigo medicine increased significantly between 2010 and 2018, by approximately 3.5 and 12.5 times, respectively. However, the use of all other drugs, except blood circulation agents, tinnitus medicine, and psychiatric agents, remained at comparable levels (Appendix A).

## 4. Discussion

This study used HIRA-NPS data from 2010–2018 to analyze the disease characteristics, treatment methods, treatment cost, type of medical institution visited, and annual distribution of patients for tinnitus. The number of patients with tinnitus, the number of claims, and the total expenditure increased each year, regardless of whether WM or KM was used. In addition, the number of patients using WM each year was 5–6 times higher than that using KM, and there was no major fluctuation between the years.

Overall, 1.4 times as many female patients as male patients sought treatment for tinnitus. The results by age were consistent with those of a previous study reporting that the number of patients with tinnitus increased with age [1]. The current results for patients using KM were similar to those of another study reporting that the use of KM services was higher among middle-aged (40–59 years) and older adults (≥60 years) in Korea, which has a dual healthcare system incorporating both WM and KM [21]. In addition, tinnitus services were found to involve outpatient rather than inpatient treatments predominantly. Moreover, almost three-quarters of WM treatments took place in clinics, as did most KM treatments.

For WM treatments, testing fees accounted for the highest percentage of total expenditures. They showed the highest increase in the average rate of change in total expenditures, indicating that greater efforts were being made to examine the condition of patients with tinnitus closely. Moreover, the rates of change in total expenditures and the total number of cases decreased for WM medication/prescription fees, which indicated a decrease in the number of cases for which drugs were prescribed for treating tinnitus. Such results are consistent with those of a previous study that reported that the use of drug therapy is not recommended for treating tinnitus [26].

For KM treatment, injection fees accounted for the highest proportion of average annual total expenditures, followed by consultation fees, medication/prescription fees, and testing fees. Injection fees included fees for acupuncture, moxibustion, and cupping, which may explain why they accounted for such a substantial portion of the total expenditures and the number of cases. Previous studies indicate that acupuncture therapy effectively reduces pain and improves the quality of life of patients with tinnitus [27,28,29]. For medication/prescription fees, the rate of change increased sharply for KM but not for WM. According to HIRA drug benefit claims, drug costs for herbal KM covered by health insurance increased from 14.2 billion won in 2010 to 35.8 billion won in 2018, while the percentage of expenditures for herbal medicine among all KM treatment expenditures also increased from 0.84% in 2010 to 1.32% in 2018. This finding is consistent with the increasing trend in insurance benefits paid for herbal medicine [30]. However, testing fees showed a decreasing trend in KM, attributable to the nature of the dual healthcare system in Korea that allows precision testing to be administered in WM only. Finally, inpatient treatment expenditures for both WM and KM were equal, whereas the average annual total expenditure for hospitalization fees increased each year.

The drugs most often prescribed for tinnitus consisted of blood circulation agents, consistent with the results from previous studies reporting that tinnitus is associated with CVD [6,31]. The second most often prescribed drug was psychiatric agents, of which diazepam and alprazolam were the most commonly prescribed. While patients with psychiatric disease codes were excluded from the study, prescription of psychiatric agents was also prevalent in patients with chief complaints as tinnitus, implying their use in the long-term management of unspecified tinnitus. Previous studies indicate that these two drugs are clinically effective in treating tinnitus [32,33]. The third most often prescribed drug was gastrointestinal or digestive agents, which may have been prescribed not to treat tinnitus directly but to address gastrointestinal disorders that commonly occur as an adverse effect of using blood circulation and psychiatric agents [32,34].

The yearly trend in major drug prescriptions has been decreasing for blood circulation agents and increasing for tinnitus agents and vertigo agents. The main ingredient in tinnitus agents is Ginkgo biloba, which is a CAM drug that research suggests is effective in treating tinnitus [35,36]. Moreover, because Ginkgo biloba is used to treat tinnitus, it can also be classified as a vasodilator [37]. The results of previous studies indicate that vertigo is associated with tinnitus, and treating vertigo could aid in treating tinnitus [38,39,40]. Similarly, CVD is associated with vertigo [41]. Our results and those of other studies reporting an association between tinnitus and CVD suggest an increase in the prescription rate of vertigo agents. However, several studies have reported that Ginkgo biloba is not effective in treating tinnitus [15,19]. Therefore, caution should be exercised when prescribing such drugs to patients. Lastly, the decrease in the prescription of vasodilators and the increase in the prescription of Ginkgo biloba and vertigo agents may be complementary owing to the vasodilation effect of Ginkgo biloba and the association between vertigo and CVD. However, additional studies are needed to confirm this hypothesis.

This study has several strengths in providing a novel approach to tinnitus analysis. First, this study examined the latest healthcare status of patients with tinnitus based on representative data sampled from nationwide medical service usage data. Second, this study analyzed the service usage of WM and KM in Korea, which has a unique dual healthcare system. Third, this study is the first to provide valid basic data for estimating future trends in drug therapies for tinnitus.

Although the study reveals important findings, it also has several limitations. First, no data was available regarding the diagnosis and treatment of tinnitus other than the benefits claims data. The HIRA-NPS only includes data submitted to receive reimbursement from the NHIS; there is the possibility of differences in the data depending on the service provider (physician), such as with the disease code, diagnosis, and treatment details, and the actual services provided to patients. Second, the study did not include an analysis of health services not covered by HIRA. Third, the study was based on a 9-year series of cross-sectional data. Although it was possible to examine follow-up treatment administered to a patient during a one-year period, subsequent long-term follow-up was not possible.

In conclusion, this study used the HIRA-NPS data to analyze medical service usage for tinnitus in Korea over a nine-year period. By analyzing the cost of WM and KM services used by patients with tinnitus, this study provides direction for reevaluating health-related payment systems. Moreover, this study examined the frequency of drugs used for tinnitus by year, and the findings can be used as basic data for determining the future direction of tinnitus treatment. Because no comparative studies have been conducted on WM and KM services for patients with tinnitus at the national level, the findings of this study could also be used as reference data for treating and managing patients with tinnitus. Moreover, the results can aid national health policy decisions, including the formulation of health insurance benefits and budget allocation for related diseases.

## Figures and Tables

**Figure 1 healthcare-10-01547-f001:**
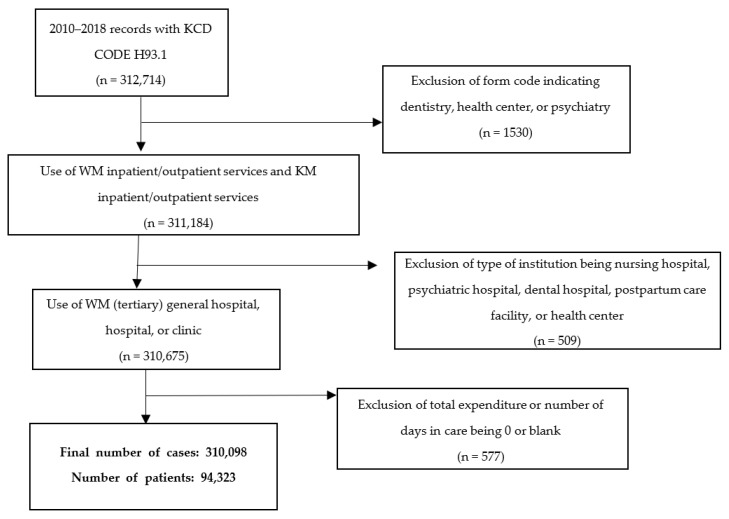
Flowchart of the study population selection process.

**Figure 2 healthcare-10-01547-f002:**
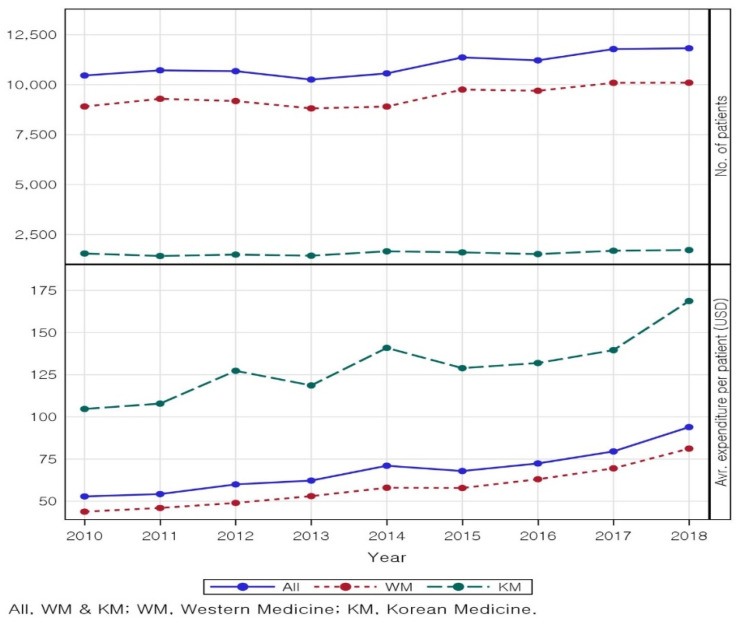
Increase in patients with Tinnitus; average expenditure per patient in WM and KM. All expenditure was converted based on the annual average exchange rate (KRW/USD), and prices were adjusted according to the health expenditure price level in 2018 (see Appendix A for details).

**Figure 3 healthcare-10-01547-f003:**
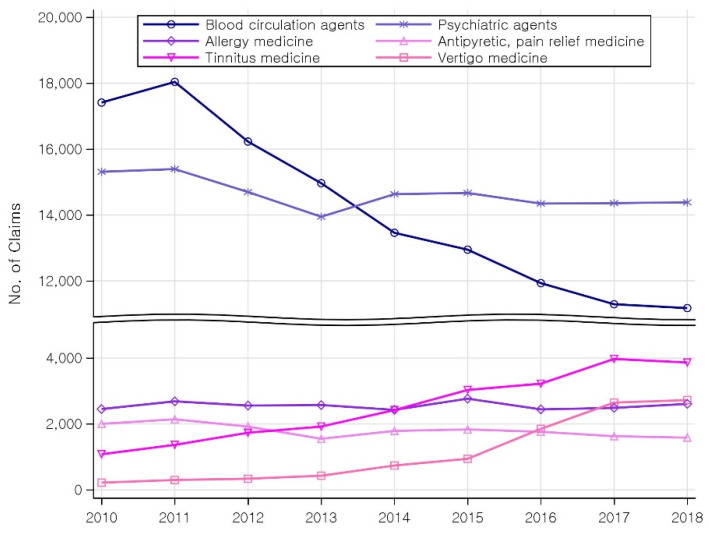
Yearly trend in major drug prescriptions.

**Table 1 healthcare-10-01547-t001:** General medical services use for patients with Tinnitus in Korea.

Year	Type of Visit	Number of Patients	Total Claims	Total Expenditure ($)
2010	Total	10,460	32,791	552,801.32
	WM	8908	21,009	390,276.54
	KM	1552	11,782	162,524.78
2011	Total	10,720	31,816	581,596.10
	WM	9293	21,964	427,641.15
	KM	1427	9852	153,954.95
2012	Total	10,678	34,924	640,470.20
	WM	9181	22,406	449,846.96
	KM	1497	12,518	190,623.24
2013	Total	10,255	31,723	638,507.68
	WM	8813	21,290	467,390.31
	KM	1442	10,433	171,117.37
2014	Total	10,565	34,588	750,577.31
	WM	8904	21,473	516,507.09
	KM	1661	13,115	234,070.22
2015	Total	11,364	35,276	771,677.49
	WM	9756	23,058	564,362.78
	KM	1608	12,218	207,314.71
2016	Total	11,218	34,728	812,234.59
	WM	9695	23,428	611,270.83
	KM	1523	11,300	200,963.76
2017	Total	11,782	36,508	937,176.46
	WM	10,093	24,020	701,422.60
	KM	1689	12,488	235,753.86
2018	Total	11,822	37,744	1,110,783.95
	WM	10,098	23,815	819,945.73
	KM	1724	13,929	290,838.22

KM, Korean Medicine; WM, Western Medicine. All expenditures were converted based on the annual average exchange rate (KRW/USD), and the price was adjusted according to the health expenditure price level in 2018 (see Appendix A for further details).

**Table 2 healthcare-10-01547-t002:** Patients’ basic characteristics.

Category		Patient							
		Total (2010–2018)		Only WM (2010–2018)		Only KM (2010–2018)		Both WM and KM (2010–2018)	
		Total N	Percent	Total N	Percent	Total N	Percent	Total N	Percent
Total number of patients		94,323	100	80,200	85.0	9582	10.2	4541	4.8
Age	<15	1784	1.89	1696	2.11	66	0.69	22	0.48
	15–24	5424	5.75	5061	6.31	243	2.54	120	2.64
	25–34	6903	7.32	6128	7.64	537	5.6	238	5.24
	35–44	10,793	11.44	9083	11.33	1163	12.14	547	12.05
	45–54	17,826	18.9	14,644	18.26	2117	22.09	1065	23.45
	55–64	21,389	22.68	17,724	22.1	2460	25.67	1205	26.54
	65–74	19,426	20.6	16,477	20.54	2003	20.9	946	20.83
	75–	10,778	11.43	9387	11.7	993	10.36	398	8.76
Gender	Male	39,495	41.87	33,659	41.97	3929	41	1907	42
	Female	54,828	58.13	46,541	58.03	5653	59	2634	58
Payer type *	NHIS	89,653	95.05	75,961	94.71	9289	96.94	4403	96.96
	Medicaid	4326	4.59	3901	4.86	293	3.06	132	2.91
	Others	344	0.36	338	0.42	-	-	6	0.13

NHIS, National Health Insurance Service. * Others include military hospitals and other medical utilities for patients under public service.

**Table 3 healthcare-10-01547-t003:** Average total expenditure, the total number of cases, and average rate of change over nine years.

	All				WM				KM			
	Total Expenditure		No. of Claims		Total Expenditure		No. of Claims		Total Expenditure		No. of Claims	
	Avr. Exp *	Avr. CR *	Avr. No	Avr. CR	Avr. Exp *	Avr. CR *	Avr. Exp *	Avr. CR *	Avr. Exp *	Avr. CR *	Avr. Exp *	Avr. CR *
Injection fee	124,039	7.9	40,907	4.5	3954	3.8	4115	1.3	120,084	8.0	36,792	4.9
Consultation fee	328,971	6.1	43,619	5.9	246,793	6.0	31,055	7.2	82,178	6.2	12,563	2.9
Testing fee	249,171	12.7	23,463	7.2	249,171	12.7	23,463	7.2	1028	(7.8)	281	(5.0)
Medication/prescription fee	8839	2.7	8720	3.2	5822	(5.4)	6076	(3.8)	3016	21.3	2644	24.9
Hospitalization fee	4439	10.5	135	7.1	3703	11.2	118	9.8	736	13.2	17	(2.7)
Treatment fee (surgery, psychotherapy, physical therapy, anesthesia, etc.)	19,692	8.8	4449	3.6	18,664	9.8	4168	4.1	-	-	-	-
Radiology fee	20,697	8.4	775	2.4	20,697	8.4	775	2.4	-	-	-	-

KM, Korean Medicine; WM, Western Medicine. * Avg. exp: Annual average gross expenditure over nine years; Avg. No.: Annual average number of cases over nine years; Avg. CR: Annual average change rate over nine years. All expenditures were converted with the annual average exchange rate (KRW/USD). The price level of health expenditure was adjusted as of the year 2018 (see Appendix A for further details).

**Table 4 healthcare-10-01547-t004:** High-frequency drug care for patients with tinnitus.

Categories	Total Claims	Average Expenditure per Claim	Average Expenditure per Patient
Blood circulation agents	127,430	3.82	11.22
Psychiatric agents	131,725	1.75	6.3
Gastrointestinal agents	88,470	2.2	5.75
Tinnitus medicine	22,612	3.48	7.59
Vertigo medicine	10,184	0.92	1.97
Respiratory agents	15,382	0.86	1.77
Antipyretic, pain reliever medication	16,209	1.1	1.98
Allergy medication	23,005	1.64	4.04
Antibacterial agent	4599	4.56	7.25
Other drugs	39,781	0.82	1.77

All expenditures were converted based on the annual average exchange rate (KRW/USD), and the price was adjusted according to the health expenditure price level in 2018 (see Appendix A for further details).

## Data Availability

The datasets generated and/or analyzed during the current study are available in the HIRA-NPS repository upon request http://opendata.hira.or.kr (accessed on 6 April 2021) and upon payment of a data request fee (300,000 KRW per dataset).

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
