# Peer review of "Trends of Medical Service Utilization for Tinnitus: Analysis Using 2010–2018 Health Insurance Review and Assessment Service National Patient Sample Data"

_healthcare, 2022, doi:10.3390/healthcare10081547_

Round 1

Reviewer 1 Report

An interesting manuscript dealing with the economic aspects of tinnitus, epidemiology, and treatment trends.

The manuscript is in general well written and of interest to the readers of “Healthcare”.

A few comments that may improve this work:

 1.       Line 54: “Insurance Statistical Yearbook of Korea, 280,389 patients were treated for tinnitus (ICD 55 code: H93.1), and the total estimated expenditure was 14.3 billion KRW, which then in-56 creased to 325,466 patients and 28.7 billion KRW in 2018 (11).” Please add the cost per patients and an estimation of this mount in $ or € thus readers can compare with the cost per patients mentioned for other countries.

2.       It is interesting that psychiatric patients were excluded from the study, however psychiatric agents were administered to the remaining patients, could the authors elaborate on this issue in the discussion? Are psychiatric patients not managed by psychiatry clinics included in the study?

3.       In table 3 when mentioning the amounts, to decimal points are too many…. In my opinion is better to avoid decimal points.

Author Response

Reviewer 1:

An interesting manuscript dealing with the economic aspects of tinnitus, epidemiology, and treatment trends.

The manuscript is in general well written and of interest to the readers of “Healthcare”.

A few comments that may improve this work:

  1. Line 54: “Insurance Statistical Yearbook of Korea, 280,389 patients were treated for tinnitus (ICD 55 code: H93.1), and the total estimated expenditure was 14.3 billion KRW, which then in-56 creased to 325,466 patients and 28.7 billion KRW in 2018 (11).” Please add the cost per patients and an estimation of this mount in $ or € thus readers can compare with the cost per patients mentioned for other countries.

- We appreciate the Reviewer’s comment. Based on the Reviewer’s comment, we revised the manuscript as follows:

In the 2010 National Health Insurance Statistical Yearbook of Korea, 280,389 patients were treated for tinnitus (ICD code: H93.1), and the total estimated expenditure was 14.3 billion KRW (12.5 million USD), which then increased to 325,466 patients and 28.7 billion KRW (25.7 million USD) in 2018.

  1. It is interesting that psychiatric patients were excluded from the study, however psychiatric agents were administered to the remaining patients, could the authors elaborate on this issue in the discussion? Are psychiatric patients not managed by psychiatry clinics included in the study?

- We appreciate the Reviewer’s comment. As the reviewer pointed out, psychiatric patients were excluded from this study. By ruling out patients diagnosed with psychiatric disorders who also have tinnitus, we were able to avoid possible confusions between psychiatric patients with tinnitus and those who suffered from tinnitus as their chief complaint. In this context, the prescription of psychiatric agents (or benzodiazepines) found in this study can be interpreted as a means of managing tinnitus and not a direct treatment targeted to treat psychiatric symptoms such as anxiety or depression.

Based on the Reviewer’s comment, we revised the manuscript as follows:

The second most-often prescribed drug was psychiatric agents, of which diazepam and alprazolam were the most commonly prescribed. While patients with psychiatric disease codes were excluded from the study, prescription of psychiatric agents were also prevalent in patients with chief complaints as tinnitus, implying their use in long-term management of unspecified tinnitus. Previous studies indicate that these two drugs are clinically effective in treating tinnitus.

  1. In table 3 when mentioning the amounts, to decimal points are too many…. In my opinion is better to avoid decimal points.

- We appreciate the Reviewer’s comment. We rounded up the averages of total expenditures and number of claims to integers and annual average change rate to one decimal place. 

Reviewer 2 Report

1) In my opinion should be made a better explanation about WM and KM.

2) It was also to be explained what the relationship of tinnitus with deafness.

3) Rectify:

Lines 125, 150, 206 and 218: Table S1

Line 172: Table S2

Line 192: Table S3

Line 204: Table S4

Line 217: Figure S1

Author Response

1) In my opinion should be made a better explanation about WM and KM.

- We appreciate the Reviewer’s comment. Based on the reviewer’s comment, we revised the manuscript and added the following paragraph:

In Korea, acupuncture and herbal medications are practiced and prescribed by Doctors of Korean Medicine as the dual healthcare system divides the medical practice sectors into Western Medicine (WM) and Korean Medicine (KM). For musculoskeletal disorders and diseases with unspecified causes, KM treatments can be an option22. For tinnitus, however, the exact utilization of both WM and KM has not yet been identified.

2) It was also to be explained what the relationship of tinnitus with deafness.

- We appreciate the Reviewer’s comment. Based on the reviewer’s comment, we revised the manuscript and added the following sentence:

Tinnitus is reported to be linked with hearing loss, although the mechanisms underlying its association are unknown3.

3) Rectify:

Lines 125, 150, 206 and 218: Table S1

Line 172: Table S2

Line 192: Table S3

Line 204: Table S4

Line 217: Figure S1

 - We appreciate the Reviewer’s comment. Based on the reviewer’s comment, we revised the manuscript as below:

All expenditures were converted using the currency exchange rate (KRW to USD) for the applicable year and adjusted based on the health sector consumer price index for 2018 (Supplementary Table 1).

Overall, 99.92% of medical services were outpatient treatments and 0.08% were inpatient treatments. WM treatments took place mostly in primary care centers, with treatments at clinics accounting for 74.17% of WM services. KM treatments were more likely to take place in primary care centers, with treatments at clinics accounting for 96.12% of KM services (Supplementary Table 2).

Considering that the average expenditure per case was $2.50, approximately 14 blood tests and/or urinalyses were performed per year (Supplementary Table 3).

Among KM services, acupuncture therapy was performed an average of 15 times per patient per year, and all other treatments were performed an average of six times per patient per year (Supplementary Table 4).

However, the use of all other drugs, except blood circulation agents, tinnitus medicine, and psychiatric agents, remained at comparable levels (Supplementary Figure 1).

Reviewer 3 Report

General comments

The title of the manuscript very well identifies its purpose. In any case, I would add that this is a study for Korea.

The subject under analysis is, in my opinion, suitable for the Journal. It is also a relevant issue, particularly in terms of Public Health and Health Economics, as evidenced by the prevalence rate of Tinnitus and the associated psycho-socio-economic costs.

In methodological terms, the manuscript is essentially descriptive and simple.

It is also a well-structured, and complete, manuscript.

Specific comments

I am of the opinion that when a manuscript is read (for the first time) without interruption, allowing the reader to clearly apprehend the objectives (if appropriate and relevant) of the study and what its results (if correct and obtained through a suitable methodology) are, it is almost always synonymous that it is a manuscript that deserves to be published. This was the case of this manuscript, which I really enjoyed reading.

In fact, being simple, from a methodological point of view, one could use (more) sophisticated analysis (e.g., time series models, Hodrick-Prescott filters, etc.) in order to identify the trend in the variables, but I do believe that this would not add much, taking into account the achievement of the study’s objectives.

That said, I have a doubt and a statement, which I recommend the authors elaborate a bit on.

Regarding the doubt, can the authors verify that what is said in the sentence “Table 4 shows the frequency of drugs used for tinnitus. Among the therapeutics used for the treatment of tinnitus, blood circulation agents were used in the highest number of cases (n=125,462), followed by psychiatric agents (n=95,349) and gastrointestinal agents (n=86,873).” (page 7: 208-210) is in agreement with the numbers that, in fact, appear in Table 4?

With regard to the statement, reading (some of) the literature on Tinnitus (and also from my own experience), I am of the opinion that Tinnitus should no longer be seen as a mere symptom or consequence of other pathologies, as it still happens, and should be seen as a disease per se. Based on their knowledge and on what this manuscript shows, do the authors agree?

Author Response

  1. Regarding the doubt, can the authors verify that what is said in the sentence “Table 4 shows the frequency of drugs used for tinnitus. Among the therapeutics used for the treatment of tinnitus, blood circulation agents were used in the highest number of cases (n=125,462), followed by psychiatric agents (n=95,349) and gastrointestinal agents (n=86,873).” (page 7: 208-210) is in agreement with the numbers that, in fact, appear in Table 4?

- We appreciate the Reviewer’s valuable comment. The numbers inserted in the main text were from the previous analyses that were found to be erroneous. Based on the Reviewer’s comment, we revised the manuscript as the following:

Table 4 shows the frequency of drugs used for tinnitus. Among the therapeutics used for the treatment of tinnitus, blood circulation agents were used in the highest number of cases (n=127,430), followed by psychiatric agents (n=131,725) and gastrointestinal agents (n=88,470).

  1. With regard to the statement, reading (some of) the literature on Tinnitus (and also from my own experience), I am of the opinion that Tinnitus should no longer be seen as a mere symptom or consequence of other pathologies, as it still happens, and should be seen as a disease per se. Based on their knowledge and on what this manuscript shows, do the authors agree?

- We appreciate the reviewer’s comment and agree that tinnitus has characteristics of an independent disorder with a wide range of associations with different diseases. Based on the reviewer’s comment, we revised the manuscript and added the following sentences in Introduction:

Due to its unknown etiology, wide range of concomitant diseases, and its influence on the quality of life, some studies defined tinnitus as an independent disorder which require a comprehensive diagnosis through etiology and patients’ comorbidities.